# Investigating the Toxicity of Compounds Yielded by *Staphylococci* on Vero Cells

**DOI:** 10.3390/toxins14100712

**Published:** 2022-10-18

**Authors:** Margaret Selina Modimola, Ezekiel Green, Patrick Njobeh, Jeremiah Senabe, Gerda Fouche, Lyndy McGaw, Sanah Malomile Nkadimeng, Kgama Mathiba, Julian Mthombeni

**Affiliations:** 1Department of Biomedical Sciences, The University of Johannesburg, Doornfontein Campus, Johannesburg 2028, South Africa; 2Department of Biotechnology and Food Technology, The University of Johannesburg, Doornfontein Campus, Johannesburg 2028, South Africa; 3Council for Scientific and Industrial Research (CSIR), Meiring Naude Rd, Brummeria, Pretoria 0184, South Africa; 4Paraclinical Sciences Department, University of Pretoria, Onderstepoort, Pretoria 0110, South Africa

**Keywords:** Vero cells, cytotoxic, fluoranthene, methyl palmitate, secondary metabolites

## Abstract

Bacterial secondary metabolites play a major role in the alleviation of diseases; however, the cytotoxicity of other metabolites cannot be ignored as such metabolites could be detrimental to human cells. Three *Staphylococci* strains *Staphylococcus aureus*, *staphylococcus epidermidis* and *staphylococcus saprophyticus* were used in the experiments. These strains are well known to cause hospital and community-acquired infections. Secondary metabolites from *S. aureus* isolated from milk of cows with clinical features of mastitis (swollen udders and the production of watery clotted milk), S. saprophyticus (ATCC 35552), and S. epidermidis (ATCC 51625) were exposed to a minimal medium then screened using Gas Chromatography High-Resolution Time-of-flight Mass Spectrometry (GC-HRTOF-MS) and identified with Nuclear Magnetic Resonance (NMR). From *S. epidermidis*, two compounds were isolated: oleamide and methyl palmitate; three from *S. aureus,* including fluoranthene, 3-methyl-2-phenyl-1H-pyrrole, and cyclo(L-Leu-L-Propyl); while *S. saprophyticus* yielded succinic acid, 1,2,6-hexantriol, veratramine, and 4-methyl-pentyl-amine. The secondary metabolites were tested for cytotoxicity using the Vero cell line. Fluoranthene exhibited toxicity with an LC_50_ of 0.0167 mg/mL to Vero cells, while the other metabolites did not. Methyl palmitate was the least toxic of all of the metabolites. The results imply that none of the compounds, except fluoranthene, pose any danger to human cells.

## 1. Introduction

Cytotoxic investigations evaluate the effect and, consequently, the fate of a compound. These investigations play a role in eliminating possible pharmacotherapy compounds that could be detrimental to host cells [1].

The investigation of the toxicity of compounds from *Staphylococcus* species [(*S. saprophyticus* (ATCC 35552), *S. epidermidis* (ATCC 51625), and *S. aureus* (isolated from milk)] could influence the decision on whether or not such compounds could be employed to alleviate challenging medical conditions. The selection of *Staphylococci* strains was due to the role these bacteria have as the cause of nosocomial infections. The ability to cause infection is attributed to the toxin or enzyme produced by these strains during host infection [2,3].

Secondary metabolism in bacteria occurs during the stationary phase, wherein bacterial species divert from biomass production to the production of small, low molecular weight bio-active molecules known as secondary metabolites [4]. Metabolite yield can be influenced by whether a bacterial culture is axenic or mixed (co-cultivation). There is a better secondary metabolite yield in mixed than axenic culture [5,6]. The co-cultivation approach could also produce higher levels of desired products [7,8]. Metabolite production also depends on temperature, pH levels, and the composition of the growth medium as well as the incubation period, [9,10]. 

Some bacterial metabolites are fundamental in the production of health-protecting products. For example, tetanus toxoid (TT), a vaccine that prevents tetanus (a condition characterized by lockjaw) was derived from *Clostridium tetani* toxins [11]. Streptomycin, a broad-spectrum antibiotic, is a product of the *Streptococcus griseus* toxin [12,13]. Over and above this, secondary metabolites can be used medically as adjuvants to enhance the activity of treatment therapies as well as weapons to target other microbes [14,15,16]. 

However, not all secondary metabolites contribute positively to health, as some are associated with health-threatening conditions. Some bacterial metabolites may be carcinogenic [17], cytotoxic [18], nephron-toxic and cardio-toxic [19]. In this study, we evaluated the possible toxic effect of bacterial secondary metabolites from *Staphylococcus* species (*S. aureus, S. epidermidis,* and *S. saprophyticus*) on Vero cells. Cytotoxicity investigations play a role in the decision as to whether a compound can be used as an antimicrobial or not [20]. These investigations contribute toward the elimination of compounds that could have a negative effect as pharmacotherapy agents [21,22].

Cytotoxicity studies can be carried out by employing Vero cells and the 3-(4,5-dimethylthiazol-2-yl)-2,5-diphenyl tetrazolium bromide (MTT) assay. Vero cells originate from African green monkey kidneys and are well recognized in research due to their accessibility and versatile nature and have been employed in various research studies, for example in plant extract and bacterial cytotoxicity investigations [23]. Vero cell lineage is continuous and has an abnormal number of chromosomes, i.e., less or more than 23 pairs of chromosomes. This characteristic allows them to be replicated through numerous cycles of division without any deterioration of function. Unlike normal mammalian cells, Vero cells are interferon-deficient, thus they do not secrete interferon when infected with viruses. However, they have the interferon-alpha/beta receptor, therefore respond normally when recombinant interferon is added to culture media.

In [24], the cellular response and mechanism of the toxic action of three different continuous cell lines used routinely in toxicological studies (HeLa, 3T3, and Vero cell lines) against rotenone (ROT, CAS 83-79-4) and Pentachlorophenol (PCP, CAS 87-86-5), known environmental pollutants was investigated. It was discovered that Vero cells superseded 3T3 and HeLa cell lines in sensitivity after PCP and ROT treatments. The detection limits in Vero cells after PCP exposure showed a significant difference from a minimum concentration of 1 micromolar (µM) when using the MTT reduction test

The MTT assay is a colorimetric test that evaluates the activity of the mitochondria, sausage-shaped membrane-bound organelles found within bacterial cells. These organelles maintain cellular homeostasis by regulating calcium signaling, apoptosis, energy production, and cell metabolism [25]. The mitochondria house the respiratory complex known as Succinate dehydrogenase (SDH). SDH catalyzes the production of nicotinamide adenine dinucleotide (NADH), a coenzyme found in living cells. Interaction between NADH and MTT results in the reduction of MTT to purple formazan. Only live cells produce NADH; therefore, the presence of purple color indicates active metabolism within bacterial cells or live cells [1]. The reduction of MTT is measured as absorbance on a microplate reader at a wavelength of 570 nm.

The study aimed to generate secondary metabolites from the aforementioned bacterial strains, identify the compounds and investigate the toxicity of the compounds on Vero cells. This exercise is necessary to evaluate if the identified compounds could be safely used as therapeutic products.

## 2. Results 

### 2.1. Laboratory Identification/Confirmation of Staphylococci Strains

The results of the three *staphylococci* strains are represented in Table 1. After incubation, the *S. aureus* strain exhibited growth on MSA agar with yellow colonies while the other two strains produced pink colonies. All the strains grew on DNA agar plates, but the clearing zones were absent around the growth of *S. epidermidis* and *S. saprophyticus* after the plate was flooded with hydrochloric acid post incubation. Coagulase production was absent in *S. epidermidis* and *S. saprophyticus,* while *S. aureus* produced coagulase. 

### 2.2. Antibiotics Susceptibility Results

The antibiotic susceptibility results are depicted in Table 2. The table shows three *staphylococci* strains and the antibiotic concentrations that these strains were tested against as well as the resistant, intermediate and susceptibility zones in millimeters. The *S. aureus* and *S. saprophyticus* strains were susceptible to all of the antibiotics allotted in Table 3, while *S. epidermidis* was resistant to Vancomycin and Oxacillin. The results imply that the strains were not multidrug-resistant strains.

### 2.3. The GC-HRTOF-MS Screening Results

The extracts from the three *Staphylococci* strains (*S. aureus* (isolated from milk), *S. epidermidis* (ATCC 51625) and *S. saprophyticus* (ATCC 35552)) were screened with GC-HRTOF-MS and are represented in Table 3, Table 4 and Table 5. Each table describes the compounds from each bacterial strain, various times for compound elution, the mass-to-charge ratio (*m/z*), the molecular mass of the compounds and the peak area (the amount of compound present). Secondary metabolites yielded by *Staphylococci* strains (Table 3, Table 4 and Table 5) were acids, alcohols, alkenes, amines, heterocyclic compounds, esters, and fatty acids. Similar compounds eluted by *S. aureus* and *S. saprophyticus* was succinic acid that was eluted at 25:44 min; fragments *(*66.5316; 69.1228) and (44.0497; 74.0237) with average peaks of 1,152,910 and 748,584 for *S. aureus* and *S. saprophyticus,* respectively.

An organosilicon compound such as dimethoxy-diphenyl silane was present in *S. epidermidis* and *S. aureus.* This compound appeared at 13:53 and 30:17 min with fragments (137,0420;167,0525) and (91.0544;167.0525) at peak averages of 957,364 and 55,316 in *S. aureus* and *S. epidermidis,* respectively.

Anticholigenic agents and halogenated pyrroles were reported only in *S. epidermidis.* The sulphides were not eluted *by S. epidermidis* and *S. aureus*, but by S*. saprophyticus* only.

#### 2.3.1. Prominent Metabolites

The prominent secondary metabolites (above 1%) yielded by *Staphylococci* are summarized in Figure 1. Data indicates that *S. saprophyticus* (ATCC 35552) yielded 9% and 12% more other compounds than *S. epidermidis* and *S. aureus*, respectively. Organosilicon compounds were more prevalent in *S. epidermidis* but ≤1% in *S. aureus* and *S. saprophyticus* while aromatic heterocyclic compounds contributed 4.87%; however, amino acids were ≤1% compared to *S. aureus* and *S. epidermidis*. Other prevalent compounds in *S. saprophyticus* were alcohols acids, alkaloids, and alkenes.

#### 2.3.2. Statistical Analysis for Metabolites Yield

Prominent secondary metabolites were distributed as per peak areas. There were seven values for *S. aureus*, five for *S. epidermidis* and eleven for *S. saprophyticus* resulting in a total number of 23. Due to the distribution (normal or not normal) of the secondary metabolites, the Kruskal Wallis test (a non-parametric test) was carried out to establish if the percentage peak area average differed significantly across the prominent metabolites yielded by the three *Staphylococci* strains, as the total sum of the values was 23. The Kruskal-Wallis results are depicted in Table 6.

In Table 6, the percentage peak area average does not differ significantly (*p* = 0.396) across the three metabolites classes as indicated by the significance level.

### 2.4. Identification of Compounds

The compounds from *S. aureus* (isolated from milk*)*, *S. saprophyticus (*ATCC 35552) and *S. epidermidis (*ATCC 51625) were reported as chemical shifts (^1^H NMR). The compounds identified in *S. aureus* were characterized by ^1^H NMR showing the signals of arrangements of aromatic hydrocarbons (Fluoranthene), heterocyclic compounds (3-Methyl-2-phenyl-1H-pyrrole) and amino acids (Cyclo(L-Leu-L-Propyl). For *S. epidermidis,* the compounds were identified as Oleamide (amide) and Methylpalmitate (ester), while *S. saprophyticus (*ATCC 35552) yielded an amine, alkaloid, alcohol, and acid. The identified compounds were Veratramine (alkaloid); 1,2,6-Hexantriol (alcohol); Succinic acid (acid) and 4-Methyl-pentyl-amine (amine). The chemical shifts of the above-mentioned compound are indicated below.

#### 2.4.1. S. saprophyticus-Derived Compounds


4-Methyl-pentylamine (C_7_H_1_).


Classification: Amine

Chemical shift: 1H NMR (600 Varian MHz, CDCL3): δ 2.58–2.53 (m, 2 H, CH2-NH2), 2.42 (s, 3 H, NH-CH3), 1.50–1.26 (m, 8 H, 4 CH2), 0.91–0.86 (m, 3 H, CH3) ppm.
Veratramine (C_27_H_39_NO_2_)

Classification: Alkaloid

Chemical shift: 1H NMR (600 Varian MHz, CDCL3) 0.83 (3 H, d, J ¼ 7.0 Hz, 27-H), 1.15 (3 H, s, 19-H), 1.40 (3 H, d, J ¼ 7.5 Hz, 21-H), 2.11 (1 H, brs, 20-H), 2.32 (3 H, s, 18-H), 2.50 (1 H, dd, J ¼ 9.0, 4.0 Hz, 22-H), 3.27 (1 H, m, 23-H), 3.52 (1 H, m, 3-H), 5.49 (1 H, brd, J ¼ 4.0 Hz, 6-H), 6.97 (1 H, d, J ¼ 7.5 Hz, 15-H), 7.22 (1 H,d J ¼ 7.5 Hz, 16-H).
1,2,6-hexanetriol (C_6_H_14_O_3_)

Classification: Alcohol

Chemical shift: 1H NMR (600 Varian MHz, CDCL3) δ = 5.37-4.56 (brs, 3H), 3.78-3.69 (m, 1H), 3.62-3.54 (m, 1H), 3.51-3.40 (m, 2H), 1.69-1.53 (m, 2H), 1.53-1.31 (m, 2H), 1.18 (d, J = 6.3 Hz, 3H) ppm.
Succinic acid (C_4_H_6_O_4_)

Classification: Acid

Chemical shift: 1H NMR (600 Varian MHz, CDCL3) δ 2.42 (α, β-CH).

#### 2.4.2. *S. aureus*-Derived Compounds


Flouranthene (C_16_H_10_)


Classification: Polycyclic aromatic hydrocarbon

Chemical shift: 1H NMR (600 Varian MHz, CDCL3): J =0.0HzH3); 8.02ppm (d, 1H, J = 8.1 Hz, H4); 8.23 ppm (d, 1H, J-7.9 Hz, H6).
Cyclo (leucyl-prolyl (C11H18).

Classification: Amino acid

Chemical shift: 1H NMR (600 Varian MHz, CDCL3) δ 0.96 (3H, d, J = 6.4 Hz, 11-CH3), 1.00 (3H, d, J = 6.4 Hz, H-12), 1.52 (1H, ddd, J = 5.0, 9.6, 14.8 Hz, H-10), 1.74 (1H, m, H-11), 1.90 (1H, m, H-4), 1.98-2.09 (2H, m, H-4, H-10), 2.13 (1H, m, H-5), 2.35 (1H, dddd, J = 3.2, 8.0, 8.0, 12.8 Hz, H-5), 3.56 (2H, m, H-3), 4.01 (1H, dd, J = 3.4, 9.6 Hz, H-9), 4.11 (1H, t, J = 8.0 Hz, H-6), and 5.86 (1H, br.s, NH).
3-Methyl-2-phenyl-1H-pyrrole (C11H11N)

Classification: Heterocyclic compounds

The NMR shift was represented as: 1H NMR (600 Varian MHz, CDCL3) ppm 7.39-7.24 (m, 4H), 7.16-7.10 (m, 1H), 6.92 (t, J = 2.5 Hz, 1H), 6.81-6.79 (m, 1H), 6.11 (t, J = 2.5 Hz, 1H), 2.10 (s, 3H).

#### 2.4.3. S. epidermidis-Derived Compounds


Oleamide (C18H35NO)


Classification: Amides

Chemical shift: 1H NMR (600 Varian MHz, CDCL3): 6.00 (br s, 1H, NH), 5.27 (m, 2H, H-9, H-10), 3.26 (dt, 2H, H-1′, 5.5, 6.0), 2.34 (t, 2H, H-2′, 6.0), 5.7), 2.17 (s, 6H, NCH3), 2.10 (t, 2H, H-2, 7.4), 2.00-1.90 (m, 4H, 2H-8, 2H-11), 1.62-1.53 (m, 2H, H-3), 1.23-1.20 (m, 20H), 0.81 (t, 3H, H-18, 6.6).
Methyl palmitate (C17H34O2)

Classification: Esters

Chemical shift: 1H NMR (600 Varian MHz, CDCL3): 5.74 (br, 1H, NH, 3.26 (t, 2H), 2.87–2.31 (m, 3H), 1.78–1.59 (m, 2H), 1.60–1.40 (br, 4H), 1.35–1.15 (br, 48H, CH2), 0.87 (t, 6H, CH3).

### 2.5. Cytotoxic Studies

Table 7 displays the LC_50_ of compounds derived from *S. aureus*, *S. epidermidis*, and *S. saprophyticus.* The LC_50_ values ranged from 0.0167–0.0441 mg/mL. Fluoranthene was the most toxic because its LC_50_ value (0.016 mg/mL) was the lowest of all the compounds, and relatively close to that of the positive control (doxorubicin) with an LC_50_ of 0.0097 mg/mL. Other compounds had relatively low cytotoxicity to Vero cell lines.

#### Statistical Analysis for Cell Viability

The ANOVA test was carried out to check the relationship between concentration and percent cell viability (Table 8 and Table 9). Table 8 shows that there is a statistically significant (*p* < 0.05) linear relationship between concentration and percent cell viability. In Table 7 the R square value (coefficient of determination) is 0.724 suggesting that 72.4% of the variance in percent viability is explained by the concentration of the sample.

## 3. Discussions

*S. aureus* strains exhibited susceptibility to Augmentin and Oxacillin. The efficacy of Augmentin against *S. aureus* is also documented by [26]. Since the *S. aureus* strain was susceptible to Augmentin and Oxacillin, it is classified as methicillin-sensitive *S. aureus* [MSSA] [27]. Such was likely to be susceptible to Ceftriaxone as shown in the current study. Ceftriaxone, a β-lactam cephalosporin, binds to the bacterial-penicillin binding proteins, disrupting the synthesis of the bacterial cell wall [28]. This antibiotic is used as an alternative to Cephazolin in the treatment of methicillin-sensitive *S. aureus*-related infections. It is a preferred antibiotic due to its shorter administration period and frequency [29]. *S. aureus* also showed susceptibility to Cotrimoxazole, contrary to what was reported in [30], where resistance to Cotrimoxazole was evident in MSSA. However, the susceptibility of *S. aureus* to Vancomycin correlates with the findings of [30]. *Staphylococcus saprophyticus* was sensitive to all the antibiotics, while *S. epidermidis* was resistance to Vancomycin *and* Oxacillin but susceptible to other antibiotics. The acceptance of *S. epidermidis* as a pathogen in various areas of the human body is progressively increasing [31] The susceptibility of stains to almost all of the allotted antibiotics imply that the strains were not multidrug resistant, therefore exposure of such strains to a nutrient depleted environment, such as minimal medium, could enhance the results regarding the production of secondary metabolites.

The GC-HRTOF-MS analyses of secondary metabolites from three *Staphylococci* strains yielded acids, alcohols, alkenes, amines, heterocyclic compounds, esters, and fatty acids. The presence of alkenes, alcohols and acids in *S. aureus* was also reported by [32,33].

Common compounds for *S. aureus* and *S. saprophyticus* include succinic acid, which was eluted at 25:44 min with fragments *(*66.5316; 69.1228) and (44.0497; 74.0237) with average peaks of 1,152,910 and 748,584 for *S. aureus* and *S. saprophyticus,* respectively. Another compound eluted was an organosilicon, dimethoxy-diphenylsilane, that appeared at 13:53 and 30:17 min with fragments (137,0420;167,0525) and (91.0544;167.0525) with a peak average of 957,364 and 55,316 in *S. aureus* and *S. epidermidis,* respectively.

The *S. saprophyticus* metabolite yield exceeded that of *S. epidermidis* and *S. aureus* by 9 and 12%, respectively. Organosilicon compounds were more prevalent in *S. epidermidis* but ≤1% in *S. aureus* and *S. saprophyticus.* In *S. saprophyticus,* aromatic heterocyclic compounds were 4.87%; however, amino acids were ≤1% compared to *S. aureus* and *S. epidermidis*. Other prevalent compounds in *S. saprophyticus* were alcohols acids, alkaloids, and alkenes.

The GC-HRTOF-MS screening revealed the presence of different compounds of which some are known to possess antimicrobial properties. The screening also revealed that there was more production of secondary metabolites in *S. saprophyticus*, which exceeded that of *S. aureus* and *S. epidermidis*. However, the abundance, time of elution and distribution of each metabolite differed from one strain to another. This was noted at the elution of Succinic acid, for *S. aureus* and *S. saprophyticus;* this acid was eluted at the same time (25:44 min); with fragments *(*66.5316; 69.1228) and (44.0497; 74.0237) and average peaks of 1,152,910 and 748,584 for *S. aureus* and *S. saprophyticus,* respectively. Another compound eluted was an organosilicon, Dimethoxydiphenylsilane in *S. aureus* with elution time being 13:53 min, fragments (137,0420;167,0525) and peak average 957364, while in *S. epidermidis,* it was eluted at 30:17min, fragments (91.0544;167.0525) and peak average 55316.

In prominent secondary metabolites, the percentage peak area average was not normally distributed. The Kruskal Wallis Test was carried out to check if the percentage peak area average differed significantly across the prominent metabolites yielded by the three *Staphylococci strains*. The Kruskal-Wallis test revealed the percentage peak area average was statistically significant (*p* = 0.396).

Extracts from *Staphylococci* were purified and, thereafter, identified by Nuclear Magnetic Resonance (NMR) using Varian 600 MHz. The identification of compounds was achieved through the analyses of the protons. Identification is necessary to know the type of compound involved, thus allowing pharmaceutical scientists to produce new health-protecting products. Proton analyses were reported as chemical shifts (^1^H NMR) with ^1^H NMR showing signals of arrangements of the chemical shifts present in *S. saprophyticus (*ATCC 35552) representing amine, alkaloid hydrocarbon, alcohol, and acid. The identified compounds were Veratramine (alkaloid); 1,2,6-Hexantriol (alcohol); Succinic acid (acid) and 4-Methyl-pentyl-amine (amine).

The cytotoxicity results revealed that none of the compounds from *S. saprophyticus* and *S. epidermidis,* and only some of the compounds from *S. aureus,* showed cytotoxicity in the Vero cells. The LC_50_ of bacterial compounds was higher than that of Doxorubicin (positive control) in toxicity on Vero cells. The LC_50_ of these compounds ranged from 0.023–0.044 mg/mL.

The compounds were viewed as being toxic if the LC_50_ was equal to or lower than doxorubicin [0.0101]. The recorded LC_50_ ranges were 0.0167–0.0441 mg/mL against Vero cells); 4-Methyl-pentyl amine [0.0231]; Fluoranthene [0.0167]; Cyclo (leucyl-prolyl [0.0310]; Oleamide [0.0333]; Veratramine [0.0274]; Methyl palmitate [0.0441]; 3-methyl-2-phenyl-1H-pyrrole [0.0341]; 1,2,6-Hexanetriol [0.0333]; and Succinic acid [0.0334]. According to the American National Cancer Institution guidelines, and stated by other researchers, compounds with a LC_50_ ≤ 20 µg/mL are harmful [34,35]. The secondary metabolites that yielded these compounds were extracted with dichloromethane and the concentration of compounds was 2000 µg/mL. According to [36] the dichloromethane-extracted compounds exhibited action on cells at the lower concentration of 573.6 µg/mL [20]. Therefore, since we used dichloromethane in the extraction process, the compounds could have an impact on Vero cells at concentrations lower than 2000 µg/mL.

The Spearman’s correlation analysis showed that the test values were statistically significant; (*p*-value < 0.05) and correlation; (r = −0.912), suggesting that there was a correlation between the concentration and percent viability. These results suggested that cell viability was dependent on the higher LC_50_ value.

Compounds to be used as therapeutic agents must be efficient at lower concentrations while non-toxic to the host cell. This selective toxicity compares the therapeutic effect of the therapeutic agent to the amount that causes toxicity. The concentration of agents should destroy pathogens but also be tolerated by the host [12].

Fluoranthene was identified as being cytotoxic with an LC_50_ of 0.0167. Fluoranthenes are classified as polycyclic aromatic hydrocarbons (PAHs). PAHs are light yellow, white, or colorless solid compounds [37]. PAHs are classified according to their molecular weight; the high-molecular-weight PAHs consist of more aromatic rings while lower molecular weight PAHs have two or three aromatic rings. PAHs are unmanageable, toxic pollutants existing in the highest concentrations [38]. They are also found in popular beverages, such as coffee and tea, due to the heating steps during preparations and atmospheric deposition on raw plants, [39]. Other sources of PAHs are man-made, such as fuels from car exhausts, diesel, and coal [40]. When PAHs are released into the environment, they are ingested or inhaled and, thereafter, stored in fatty tissues, or metabolized and then excreted in urine [41]. The PAHs reduce ATP production in the mitochondrion, causing alterations in mitochondrial morphology and hindering mitochondria-dependent apoptotic pathways [42].

Fluoranthenes are the most toxic among the PAHs. The toxicity of fluoranthene was described by [43] wherein exposure to fluoranthene for 24 h negatively affected the photosynthetic ability of seven species of marine algae. The same species decreased in cell density after 72 h of exposure.

The LC_50_ of the other bacterial metabolites was higher than that of the positive control so they were less toxic to Vero cells. Furthermore, methyl palmitate was considered the least toxic of the nine compounds because its LC_50_ value (0.04 mg/mL) was furthest from the control. Other identified compounds are discussed below.

Succinic acid is produced aerobically and anaerobically by most bacteria as by-products of metabolism. This acid controls the growth of various bacterial species and possesses different levels of efficiency as an antimicrobial [44]. Succinate, a salt from succinic acid, plays a role in the production of fumaric acid necessary for initiating Krebs’s cycle, an energy generation mechanism important for normal body functioning. Succinate supplements degrade toxic aldehydes (by-products of alcohol metabolism) to water and carbon dioxide. Succinic acid enhances the recovery of immune as well as neural systems and it is a well-recognized antibiotic due to its acidic nature; however, it is corrosive at higher concentrations [45,46].

Methyl palmitate, a fatty acid methyl ester (FAME) plays a role as an antioxidant and antimicrobial. It was reported by [47] that the antimicrobial properties of microalgae were due to the higher concentration (23.08%) of palmitic acid. FAMEs have strong antimicrobial activities at the lowest MIC. This result indicated that FAME possesses antioxidant as well as antimicrobial properties against health-threatening conditions. Other identified non-toxic compounds include the following.

The antibacterial and antifungal properties of Cyclo (leucyl-prolyl) against bacteria and pathogenic fungi were reported by [48] and [49] respectively. MMS-50 also exhibited a bacteriostatic effect on *Streptococcus mutants* at the minimum and maximum inhibitory concentrations of 100 and 250 μg/mL, respectively [50].

Methyl-2-phenyl-1H-pyrrole originates from pyrroles; pyrroles interact with biomolecules of living systems to form compounds of medicinal importance. Pyrrole-containing antibiotics are widely used medically and agriculturally and are efficient against gram-negative (*E. coli*) and Gram-positive bacteria (*S. aureus*) at 30 and 31 µg/mL, respectively. Pyrrole antibiotics include tetrapyrrole prodigiosin, chlorinated pyrroles, and aminocoumarin [51].

Oleamides, also present in endophytes, are active against disease-causing agents. They target bacterial protein synthesis and cause leakage of intracellular components. The therapeutic properties cover a range of conditions, such as diabetes, cancer, and parasitic and bacterial infections [52].

Veratramine lowers blood pressure and plays a role in basal cell carcinoma therapeutics [53]. It was also reported that the antitumor activity of veratramine that prevented the downstream signaling pathway of transcription factor activator protein-1 (AP-1), which regulates apoptosis and other cell functions [54].

Methyl-pentyl amine is also recognized for its antimicrobial activity. Co-polymers derived from 4-methyl-pentyl amine, imidized and 3-(dimethylamino) (DMAPA), and 1-propylamine showed antimicrobial properties against Gram-negative bacteria [55].

1,2,6-Hexanetriol is recognized for its non-toxic nature and is, therefore, used as a solvent base for many steroids in cream applications and skin conditioners. Hexanetriol is also used in pharmaceutical drug manufacturing and testing. It can substitute glycerol due to hygroscopicity and stability. It enhances the efficacy of ingredients used in a formulation used in crop protection. Hexanetriol derivatives are used as corrosion inhibitors. It is recognized for its stability and high boiling point [56].

## 4. Conclusions

The aim of screening secondary metabolites from *Staphylococci* and the subsequent identification of compounds resulted in nine compounds. Eighty-nine percent of the identified compounds were considered safe to Vero cells and, therefore, to human cells based on high LC_50_, except for fluoranthene.

## 5. Methods

The investigation of the cytotoxic effect of *staphylococci* compounds on Vero cells was performed in vitro to determine the effect of compounds on Vero cells and, thus, on human cells. 

### 5.1. Staphylococci Strains

The *staphylococci* strains used for the experiment are depicted in Table 10. These include *Staphylococcus aureus* (*S. aureus*), *Staphylococcus epidermidis* (*S. epidermidis*) [ATCC 51625] and *Staphylococcus saprophyticus* (*S. saprophyticus*) [ATCC 35552]. These bacterial strains were in porous beads, which served as carriers to support the micro-organisms and were stored at 2 °C. They were purchased from Thermofisher Scientific, Johannesburg, South Africa) except for the Methicillin-sensitive *Staphylococcus aureus,* which was isolated from milk. The *S. aureus* sample was obtained from cows with clinical features of mastitis such as swollen udders and the production of watery clotted milk [57]. Using an automatic somatic cell counter (SCC), the number of somatic cells (white cells)/mL of milk was enumerated. Counts of 200,000 cells/mL of milk or more were indicative of mastitis [58,59]. Five hundred (500) μL of milk from mastitic cows was centrifuged at 200× *g* for 5 min at room temperature and the sediment was cultured on blood and nutrient agar. 

#### 5.1.1. Identification and Confirmation of Staphylococci Strains and the Preservation of *S. aureus* Strain

The culture characteristics of three *staphylococci* strains (*S. epidermidis*, *S. saprophyticus* and *S. aureus*) was identified by using laboratory tests such as growth on the Mannitol salt agar (MSA) and DNAse plates as well as antimicrobial susceptibility testing. The MSA plate differentiate between mannitol-fermenters (*S. aureus*) and non-fermenters (other *staphylococci*). The Mannitol-fermenting bacteria appear as yellow conies while non-fermenters are pink. The ability to grow in a high concentration of salt (7.5%), such as in MSA plate, is a characteristic of the *staphylococcus* genus [60]. DNase agar, a differential medium that determines if bacteria produce an enzyme deoxyribonuclease (DNase). This enzyme catalyzes the hydrolytic cleavage of phosphodiester linkages in the DNA backbone, thereby degrading DNA. Bacteria that produce the enzyme DNAse hydrolyze DNA and this is indicated by a clearing zone around the colonies on the DNA agar upon flooding the plate with hydrochloric acid [61]. Coagulase is an enzyme that facilitates the conversion of fibrinogen to fibrin and it distinguishes *S. aureus* from other *staphylococci* [62]. For the coagulase test, colonies from each bacterial strain were put on a slide, emulsified, then 2 drops of plasma were added, thereafter the slides were examined for clumps. 

After identification, the *S. aureus* strain was preserved by inoculating a colony from MSA, into 500 μL nutrient broth (NB) and incubated overnight, thereafter 500 μL of the overnight culture was added to 500 μL of 50% glycerol in a 2 mL screw top tube, gently agitated, and frozen at −80 °C. The preservation of *S. aureus* strain was necessary because unlike other *staphylococci* used in this research, it was not an ATCC strain, therefore it was not priorly preserved.

#### 5.1.2. Resuscitation of Staphylococci Strains

Resuscitation of strains refers to the process of reviving bacterial strains after cryopreservation. For *S. aureus,* previously glycerol preserved strains were resuscitated by adding 500 uL of preserved culture and 500 µL of NB into 1000 µL; thereafter, incubated overnight at 37 °C. After incubation, the contents of the vial were transferred into a 500 mL Schotts bottle. For other bacterial strains, each bacteria-containing bead was reconstituted by immersing the bead into a vial of 1 mL nutrient broth and incubated for 18–24 h at 37 °C. For other *staphylococci* strains; each bacteria-containing bead was reconstituted by immersing the bead into a vial of 1 mL nutrient and incubated for 18–24 h at 37 °C. The content from each vial was thereafter transferred into the corresponding bottle of 500 mL nutrient broth and further incubated for 18–24 h at 37 °C. 

#### 5.1.3. Susceptibility of Staphylococci Strains to Antibiotics 

The susceptibility of the *Staphylococci* to antibiotics was tested to check if they possess the antibiotic-resistant characteristic. Susceptibility to antibiotics refers to the ability of antibiotics to kill or inhibit bacteria [63]. The selection of the antibiotics was according to the South African treatment regimen guidelines for *Staphylococci* infections. 

A colony from 24-h-old subcultures of each strain was inoculated into a vial containing 2 mL saline and vortexed for 2 min to obtain a uniform suspension with turbidity equivalent to 0.5 McFarland standard (1.5 × 10^8^ colony forming units (CFU/mL). The suspension of each strain was streaked onto the corresponding Muller-Hinton plates purchased from Thermofisher, Johannesburg, South Africa) and, thereafter, the discs of Augmentin (30 µg), Ceftriaxone (30 µg), Oxacillin (5 µg), (30 µg) and Cotrimoxazole (25 µg) were placed onto the aforementioned streaked plates, then incubated for 24 h [64,65]. After incubation, the susceptibility zone of each antibiotic was measured using a caliper. Susceptibility was recorded as millimeters (mm).

### 5.2. Minimum Broth Preparation and Secondary Metabolites Production 

A simulated environment was developed to enhance the bacterial strains to undergo secondary metabolism. This was achieved through the preparation of a nutrient-limited growth medium, minimal broth. The minimal broth was prepared according to [66]. The protocols involved weighing and dissolving 0.5 g NaCl, 1.0 g NH_4_Cl, 3.0 g KH_2_PO_4,_ and 12.8 g Na_2_HPO_4_ in 478 mL deionized water, which was autoclaved at 121 °C and thereafter cooled to 50 °C. When the salts were cooled, 0.1 mL (Thiamine 0.5% *v*/*w* solution), 0.1 mL of 1M CaCl_2_, 2 mL (1M of (MgSO_4_) and 20 mL (Glucose 20% solution) were then filter sterilised into an M9 salts solution. Fifty (50) mL of previously incubated broths of *S. aureus, S. epidermidis,* and *S. saprophyticus* were transferred into separate 500 mL of minimal broth and, thereafter, placed in a shaking incubator (150× *g*) for 7 days at 30 °C. After 7 days, the bacterial culture broths were centrifuged for 15 min at 10 000× *g* to remove the biomass [66].

### 5.3. Metabolites Extraction and Analyses

#### 5.3.1. Extraction

Extraction of secondary metabolites was performed according to the protocol stated by [67], however, with few modifications. Accordingly, equal volumes of dichloromethane and ethyl acetate (1:1, *v*/*v*) were used for the extraction. One hundred (100) mL each of dichloromethane and ethyl acetate (1:1, *v*/*v*) were added to 200 mL of sample, and the mixture vortexed and transferred to a separation funnel and shaken thoroughly, each time with the lid opened to release excess pressure. The separation funnel was then mounted onto a ring stand to allow the separation of the phases. 

The separation of phases occurred due to gravity and was based on the principle that immiscible liquids separate into layers depending on their densities, creating different layers of solution-solute. The Teflon stopper and the tap were then opened to release the lower phase into a clean beaker. After the removal of the lower phase, the Teflon stopper was closed, and the upper layer was poured out through the top into another container. The upper layer was concentrated using a vacuum rotary evaporator. The temperature at which the sample was to be concentrated was selected, taking into consideration the average boiling point of dichloromethane (39.6 °C) and ethyl acetate (77.1 °C). The samples (upper layers) from *S. aureus*; *S. epidermidis* and *S. saprophyticus* were therefore concentrated at 58 °C, (average of 39.6 °C and 77.1 °C); thereafter, the secondary metabolites were freeze-dried, and each was transferred to previously weighed sterile flasks and stored in a dark cupboard. 

#### 5.3.2. Screening and Analyses

The preparation for sample screening and analyses was performed according to [68]. Previously dried extracts from each *S. aureus*, *S. saprophytic,* and *S. epidermis* was individually reconstituted by adding 1 mL of chromatographic grade methanol and then filtered into amber vials. 

The GC-HRTOF-MS system (LECO Corporation, St Joseph, MI, USA) operating at a high resolution was calibrated using Perfluorotributylamine (PFTBA) and 11 masses as the pre-analysis calibration, C_5_F_10_N (*m/z* 263.9871), C_8_F_16_N (*m/z* 413.9775), C_9_F_18_N (*m/z* 463.9743), C_9_F_20_N (*m/z* 501.9711), C_3_F_6_ (m/z 149.9904), C_2_F_4_ (*m/z* 99.9936), C_2_F_4_N (113.9967), CF_3_ (*m/z* 68.9952), C_2_F_5_ (*m/z* 130.9920), and C_4_F_9_ (m/z 218.9856). The intensity and resolution were 41.392 and 40.200, respectively. A microliter (1 µL) of each previously methanol-treated sample was injected into the system using helium gas as the carrier gas. The transfer and inlet temperatures were 225 and 250 °C, respectively. The temperature of the oven was set at 70 °C, and kept for 0.5 min, thereafter, adjusted from 10 °C /min to 150 °C and retained for 2 min. The oven temperature was then adjusted from 10 °C/min to 330 °C and kept for 3 min to bake out the column. The triplicate of each sample, respectively, was introduced into the GC-HR-TOF-MS equipment with solvent blanks to check for contamination and impurities.

From the data obtained, peak selection, retention time alignment, and matching detection were carried out on the ChromaTOF-HRT^®^ software (LECO Corporation, St Joseph, MI, USA). The data were also processed by making use of other parameters, such as a signal-to-noise of 100 and a minimum match similarity of >70% before the compound name was assigned by comparing the molecular formula, retention time, and mass spectra data. Percentage peak areas were then calculated, and the respective observed *m/z* fragments obtained from the ChromaTOF-HRT^®^ data station were recorded. The metabolite class was then elucidated with the corresponding *m/z* fragments and molecular formula. The concurrent versions system (CVS) GC-MS data were converted into Excel, then all the noise peaks were deleted. Only compounds after 183 s and appearing in 2 or all were considered. The compounds were characterized based on their chemical and physical modes of action.

#### 5.3.3. Purification of Crude Secondary Metabolites and Identification of Compounds from Staphylococci 

The crude secondary metabolites from *S. aureus, S. epidermis* and *S. saprophytic* were purified using the column chromatography method to the method stated in [69], with modifications. This process aimed to achieve pure compounds. Purification of the extracts was carried out using adsorbent and mobile phases, the former comprising of silica gel and methanol while the latter consisted of aluminum, thin layer chromatography (TLC) plates, and solvents of various concentrations. For the adsorbent phase; 1.2 g was dissolved in 25 mL of methanol and then absorbed with 6 g silica gel. The sample was dried and transferred to a column packed with silica gel. The sample was then eluted with a 2, 5, 8, 10, 15 and 20% methanol and dichloromethane solution. For the mobile phase we used 5% methanol and dichloromethane (comprised of 5 and 95 mL of methanol and dichloromethane); 10% methanol and dichloromethane (10 and 90 mL of methanol and dichloromethane), and 10% acetone and dichloromethane (10 and 90 mL acetone and dichloromethane), respectively. The TLC plates were developed with the above-mentioned solvents to accommodate the polar and less polar secondary metabolites.

Ten mL fractions (parts collected from a batch of a compound during the separation process) from the column were collected in a test tube each time. This collection procedure was carried out during elution for all concentrations. After collection, each fraction was spotted on a TLC plate using a capillary tube to check for the presence of compounds. Spotting was repeated 2× or 3× to concentrate the compound onto the spot for improved compound detection. The TLC plate had a row of spots corresponding to various fractions and then eluted in tanks with 10% methanol and dichloromethane to identify and check the retention factor (RF). The RF of a compound referred to the distance of the compound divided by the distance of the solvent front. The TLC was then viewed under UV light at 254 nm wavelengths. Fractions containing pure compounds were collected and those with similar RF were combined. Such compounds were placed in a fume hood to evaporate solvents; thereafter, they were dried and stored in vials.

For the impure extracts, the solid phase extraction (SPE) method was used. This method involves a solid adsorbent found within a cartridge. A one-gram silica gel cartridge was used to elute the sample. The cartridge was conditioned by adding 10 mL each of distilled water, then dichloromethane, and distilled water again; thereafter, the impure sample was loaded 10 mL at a time. The cartridge was loaded on a vacuum pump to allow for the separation process. After collecting the sample, the cartridge was rinsed with distilled water and subsequently with 5, 10, 20, and 100% methanol concentrations to decrease polarity.

The extracts were further purified employing a 10 mL acetonitrile and then a methanol pre-conditioned Varian Bond Elute C18 cartridge. Extracts were thereafter eluted with acetonitrile and methanol at the following ratios: 20, 40, 60, 80 and 100% except for *S. saprophytic,* which was less polar, therefore the sample was eluted with acetone and dichloromethane, and thereafter with 100% acetonitrile. The collected compounds were identified using a Varian 600 MHz NMR spectrometer (Agilient Technologies, CA, USA). This spectrophotometer is a three-channel instrument that operates at a ^1^H frequency of 600 MHz (14.1 T). It has a double resonance broad-band-probe head (600 DB Auto X) for general solutions analyses while well a triple-resonance-probe [5 mm Auto HCN PFG] is for biological solutions. [7,70]. 

Before the analyses, each pure sample was dissolved in 0.7 mL deuterated chloroform [CDCl_3_] (Sigma Aldrich, Taufkirchen, Germany) and then transferred into clean NMR tubes. The height of the sample in the tube was ensured to be around 5 cm, then the tubes were loaded onto the sampler. The sampler containing the MNR tubes was then wiped off with a clean paper towel to allow for proper grip during spinning and to avoid contaminating the spinner.

The tubes were then positioned in such a way that the solution was situated in the detected region of the NMR probe of the Varian 600 MHz spectrometer. An equidistant between the center of the detected region and the meniscus of the solution at the top as well as the bottom of the NMR tube was ensured to allow for the sample to shim properly. The Varian 600 MHz spectrometer was also equipped with a shim, a device that adjusts the homogeneity of the magnetic field, thus the samples were shimmed before analyses. The data obtained were processed using Varian VNMRJ Software. 

### 5.4. Preparation, Proliferation, and Harvesting of Vero Cells 

A cryopreserved (−80 °C) vial containing Vero cells [E6 cell lines] (Thermo Fisher Scientific, Johannesburg, South Africa) was thawed by gently agitating in a 37 °C water bath, ensuring that the cap of the vial was not submerged to prevent possible contamination from the water-bath. After thawing, the vial was sprayed with 70% ethanol to maintain sterility. Vero cells suspension from the cryovial were then transferred into a 15 mL conical tube containing 10mL minimal essential medium [MEM] (Thermo Fisher Scientific, South Africa) and thereafter centrifuged 400× *g* for 5 min at room temperature. The supernatant was discarded, and the cells were re-suspended in 10 mL MEM containing 5% fetal calf serum (Biological Scientific Solutions, New Delhi, India) and 0.1% gentamicin (Virbac Pharmaceuticals, Johannesburg, South Africa), thereafter transferred into a 50 cm^2^ vented-cap tissue culture flask and incubated at 37 °C with 5% CO_2_ until proliferation was achieved.

The proliferation of Vero cells is dependent on their attachment to the solid surface. Cells were monitored every second day and the media was changed every fourth day till the cells reached a >90% confluent monolayer. Confluence refers to the state where the culture flask contains twice the number of cells compared to the initial amount. Cell confluence was also identified by the presence of a turbid appearance in the culture medium, as the cells clumped together, and a decrease in medium pH due to the production of lactic acid as the metabolism by-product.

When the culture in the flask was turbid and contained almost double the number of cells compared to the initial start-up culture, the growth medium was discarded; then 2 mL Dulbecco’s Phosphate Buffered Saline [DPBS] pH 7.2–7.4 (Thermo Fisher Scientific, Johannesburg, South Africa) was added to the cells. The cells were then centrifuged for 5 min at 400 rpm and the pellet was re-suspended in 10 mL DPBS. A 5 mL volume of Trypsin ethylenediaminetetraacetic acid (EDTA) was added to the flask containing the cells and incubated for 2–3 min at 37 °C until the cells detached from the flask. A total of 5 mL MEM was added to the culture to inactivate the trypsin EDTA; the culture was then transferred to a clean flask and the concentration of cells was adjusted to 5 × 10^4^ cells/mL using MEM [71]. 

### 5.5. Cytotoxicity Assay

A cytotoxicity assay was carried out as described initially by) [72] and later used by) [73]. The serial dilutions of compounds were prepared in MEM by pipetting two hundred (200) µL of the cell suspension into each well of columns 2 to 11 of a sterile 96-well microtiter plate. Two hundred µL of MEM was added to columns 1 and 12 to maintain humidity, minimize evaporation, and consequent well-to-well variability. The plates were then incubated in a 5% CO_2_ incubator at 37 °C for 24 h until the cells reached the exponential growth phase. The MEM was aspirated from the cells, which were then washed with 150 mL phosphate-buffered saline [PBS], (Whitehead Scientific, Johannesburg, South Africa) and replaced with 200 µL of MEM with compounds of each of *S. aureus*, *S. epidermidis,* and *S. saprophyticus* at differing concentrations in quadruplicate.

The cells were least disturbed during aspiration and the addition of the medium and test compounds, respectively. A positive control [doxorubicin chloride] (Pfizer Laboratories, Johannesburg, South Africa) and untreated cells (negative control) were included in each assay. The microtiter plates were incubated in a 5% CO_2_ incubator at 37 °C for 48 h. 

After incubation, the MEM was aspirated and the cells were washed, thereafter 30 µL of 3-(4,5-dimethylthiazol-2-yl)-2,5-diphenyltetrazolium bromide [MTT] (Sigma Aldrich, Taufkirchen, Germany) from a stock solution of 5 mg/mL in PBS was added to each well and the plates were re-incubated at 37 °C in a 5% CO_2_ incubator for 4 h. The medium from each well was then carefully removed; however, not disturbing the MTT crystals in the wells. Fifty µL was added to each well to dissolve the MTT formazan crystals with the plates gently agitated to ensure thorough mixing. 

The wells in column 1, containing MTT and medium but without cells, were used to validate the performance of the plate reader (BioTek Synergy, Winooski, VT, USA) equipped with KC4 software [1.20.0.42] (BioTek Instruments Winooski, VT, USA) data reduction software. The cell viability percentage was calculated using the following formula: Percentage (%) Viability = Sample absorbance/control absorbance) × 100.

The lethal concentrations (LC_50_) values were calculated as the concentration of compounds resulting in a 50% reduction of absorbance relative to that of untreated cells. The linear regression analysis of the concentrations-response curve plotted between the sample concentration of two inherent assays was used to obtain the 50% lethal concentration of the positive control and that of the tested compounds.

## Figures and Tables

**Figure 1 toxins-14-00712-f001:**
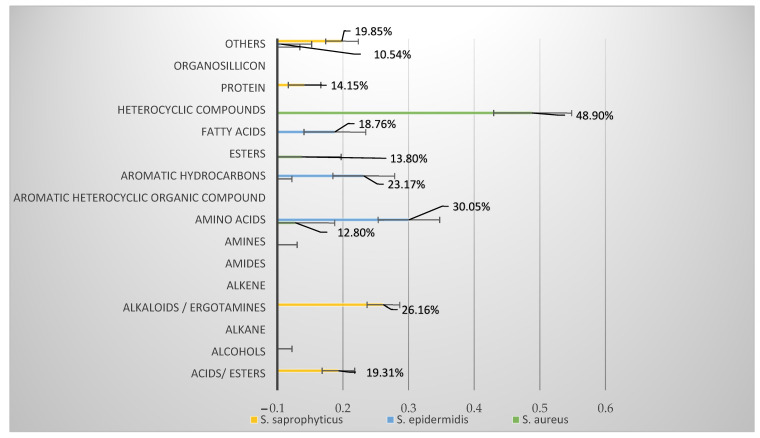
Prominent metabolites yielded by *Staphylococci*.

**Table 1 toxins-14-00712-t001:** Laboratory tests.

Bacterial Strain	Growth on MSA Plate	Growth on DNAse Plate	DNAse Production (Clearing Zone of Clearing)	Coagulase Production
*S. aureus*	yellow colonies	+	+	+
*S. epidermidis*	pink colonies	+	−	−
*S. saprophyticus*	pink colonies	+	−	−

+ = positive, − = negative.

**Table 2 toxins-14-00712-t002:** Antibiotic sensitivity results.

Bacterial Strain	Antibiotic Concentration	Res.	Int.	Sus.	Zone-Size (mm)/Results
*S. aureus* (Isolated from milk)	Augmentin (30 µg)	≤13	14–17	≥18	25 = S
	Ceftriaxone (30 µg)	≤13	14–20	≥21	20 = S
	Oxacillin (5 µg)	≤12	13–20	≥21	22 = S
	Vancomycin (30 µg)	≤10	10–11	≥12	20 = S
	Cotrimoxazole (25 µg)	*≤*10	11–15	≥16	20 = S
*S. epidermidis*	Augmentin (30 µg)	≤13	14–17	≥18	25 = S
	Ceftriaxone (30 µg)	≤13	14–20	≥21	20 = S
	Oxacillin (5 µg)	≤12	13–20	≥21	11 = R
	Vancomycin (30 µg)	≤10	10–11	≥12	8 = R
	Cotrimoxazole (25 µg	*≤*10	11–15	≥16	20 = S
*S. saprophyticus*	Augmentin (30 µg)	≤13	14–17	≥18	25 = S
	Ceftriaxone (30 µg)	≤13	14–20	≥21	20 = S
	Oxacillin (5 µg)	≤12	13–20	≥21	22 = S
	Vancomycin (30 µg)	≤10	10–11	≥12	20 = S
	Cotrimoxazole (25 µg)	*≤*10	11–15	≥16	20 = S

Sus. = susceptible, Res. = resistant, Int. = intermediate.

**Table 3 toxins-14-00712-t003:** Secondary metabolites yielded by *S. aureus* (isolated from milk).

Metabolites Class	RT (min)	Observed Iron *m/z*	*m/z*	Metabolite Name	Molecular Formula	Peak Ave
Acids	05:45	269.0349	111.0680; 131.1293	5-(2,6-dichlorophenyl)thiophene-2-carboxylic acid	C_11_H_6_Cl_2_O_2_S	532,778
	08:03	219.1047	109.0762; 70.0409	(2-bromo-4-nitrophenyl)acetic acid	C_10_H_7_F_3_O_2_	584,679
	25:44	324.1662	66.5316; 69.1228	Succinic acid	C_4_H_6_O_4_	1,152,910
Alkaloids	12:52	219.1377	57.0700; 70.0778	9-Eicosene, (E)-	C_20_H_40_	243,785
	17:03	210,1256	124,0631; 180,0893	221U 5,6,8-Indolizidine	C_15_H_27_N	2,313,600
Alkenes	23:09	265.1707	55.0545; 57.0701	1-Nonacosene	C_29_H_58_	779,656
Amides	17:50	157.1420	59.0367; 72.0444	Pelargonamide	C_9_H_19_NO	2,852,295
	19:38	264.1947	59.0367; 72.0444	Oleamide	C_18_H_35_NO	5,474,773
	03:57	101.1201	44.0497; 43.0179	Methyl isopentyl amine	C_6_H_15_N	1,176,352
Amines	04:40	260.2725	84.0808; 31.0181	Cholestan-3-amine, N,N,4,4-tetramethyl-, (3ß,5a)-	C_31_H_57_N	4,216,599
	18:08	182.0842	182.0838; 310181	1-amino-4-azafluorene	C_12_H_10_N_2_	424,629
	18:28	227,2210	59.0367; 72.0445	2-Fluoroisoproterenol	C_11_H_16_FNO_3_	440,174
	05:28	85.0412	85.0522; 43.0180	(Difluorophosphino)amine; Aminodifluorophosphine	F_2_H_2_NP	8,467,743
	11:47	163.0993	104.0621; 191.1430	Acetamide, N-phenethyl-	C_10_H_13_NO	7,464,331
	21:23	268.1933	67.1005; 77.0230	Flexzone 7L	C_18_H_24_N_2_	175,865
Amino Acids	15:45	154,0737	83,0729; 111,0678	Cyclo-prolylglycine	C_7_H_10_N_2_O_2_	24,522,536
	16:24	210,1360	70.0652; 72.0808	Cyclo(leucyloprolyl)	C_11_H_18_N_2_O_2_	75,187,945
Azoles/Thiazoles	23:02	315.1136	39.0832; 76.0183	Tinuvin 326	C_17_H_18_ClN_3_O	235,435
Azoles/Heterocyclic Compounds	12:55	157.0888	128,0622; 156,0811	3-Me-4-Ph-pyrrole	C_11_H_11_N	620,446
Biphenyl Compounds	12:57	169.0889	72.0808; 169.0887	4-Biphenylamine	C_12_H_11_N	282,773
Dicarboxylic Acids	07:14	100.0222	56.0259; 82.0651	Succinyloxide	C_4_H_4_O_3_	3,109,952
Ergot Amines/Ergot Alkaloids	21:47	273.9903	125.0708; 70.0652	Dihydroergotamine	C_33_H_37_N_5_O_5_	31,862,103
Esters	20:13	256.1935	165,1026; 235,1317	Methyl 2,2’,4-tri-O-methylanziate	C_28_H_38_O_7_	765,489
	22:18	329.0346	66.5315; 79.0293		C_24_H_28_BrNO_4_	1,654,714
	09:38	344.0148	70.1217; 72.9902	3-Methylbutyl N-heptafluorobutyryltryptophanate	C_20_H_21_F_7_N_2_O_3_	506,683
	11:53	194.0939	121,0286; 149,0601	Ethyl 4-ethoxybenzoate	C_11_H_14_O_3_	260,016
Fatty Acid Esters	17:11	228.2046	74,0363; 87,0442	Methyl tridecanoate	C_14_H_28_O_2_	586,047
Fatty acids	21:38	255.2509	59.0367; 72.0444	Palmitic amide	C_16_H_33_NO	2,419,744
Fatty Acids/Palmitic Acids	18:13	258,2503	43.0545; 102.0676	Hexadecanoic acid, isopropyl ester	C_19_H_38_O_2_	357,808
Fatty Amides	18:33	171.1621	59.0367; 72.0445	Capramide	C_10_H_21_NO	435,229
	19:49	199.1895	59.0366; 72.0444	Lauramide	C_12_H_25_NO	15,918,923
Fluorenes/Aromatic Hydrocarbons	19:24	202.0777	100.0306; 202.0777	Fluoranthrene	C_16_H_10_	106,095
Furans	15:40	128.0580	44.0496; 128,0580	Methyl 5-methylfuryl sulfide	C_6_H_8_OS	3,347,846
Heterocyclic compounds	08:18	117.0574	90.0465; 117.0574	Ketole	C_8_H_7_N	3,834,426
	08:25	267.9995	84.0808; 105.0700	Pyridrol	C_18_H_21_NO	25,271,632
	10:54	155,0730	155.0730; 127.0543	3-PhenyIpyridine	C_11_H_9_N	1,187,944
	12:40	219.0385	70.0652; 97.0888	Aprobarbital	C_10_H_14_N_2_O_3_	14,490,692
	19:40	254.2011	89.0391; 171.0919	Fenoharman	C_18_H_18_N_2_	588,461
Pyridones	23:51	227.1449	67.0102; 75.5266	1-(2-Acetyl-3-methylphenyl)-2(1H)-pyridinon	C_14_H_13_NO_2_	503,418
Hydroxytryptophan/Hydroxy Amino Acid	11:50	219.1475	146.0967; 130.9917	4-Hydroxy-DL-tryptophan	C_11_H_12_N_2_O_3_	143,938
Nitriles	19:00	139.0868	139.0868; 198.1152	3,5-difluoro-benzonitrile	C_7_H_3_F_2_N	191,900
Organosilicon compounds	13:58	418.0349	73.0468; 73.0468	Hexadecamethylcyclooctasiloxane	C_16_H_48_O_8_Si_8_	658,716
	13:53	244.0919	137.0420; 167.0525	Diphenyldimethoxysilane	C_14_H_16_O_2_Si	957,364
	11:15	505.1063	73.0468; 147.0657	CTK6B0391	C_18_H_52_O_7_Si_7_	1,255,677
Phenols	23:41	209,1376	190.0977; 135.0554	Cinnamolaurine	C_18_H_19_NO_3_	581,769
	18:51	225.0899	93.0574; 225.0896	2-(benzotriazol-2-yl)-5-methylphenol	C_13_H_11_N_3_O	299,376
Phenols/Organic Hydroxy Compound	12:36	206.1665	191.1430; 163.1119	Phenol, 2,5-di-tert-butyl-	C_14_H_22_O	148,266
Thiazole	06:50	140.1421	125.0836; 91.0254	4-(Trimethylsilyl)pyrazole	C_6_H_12_N_2_Si	3,059,569
	13:15	181.0014	68.6765; 108.0031	Benzothiazole, 2-(methylthio)	C_8_H_7_NS_2_	70,845
	07:20	135.0138	69.1227; 72.0686	benzisothiazole	C_7_H_5_NS	164,406
Unsaturated Aliphatic hydrocarbons	17:53	282.0523	57.0700; 69.0699	(5E)-5-Icosene	C_20_H_40_	1,126,073

**Table 4 toxins-14-00712-t004:** Secondary metabolites yielded by *S. epidermidis* (ATCC 51625).

Metabolites Class	RT (min)	Observed Ion *m/z*	*m/z*	Metabolite Name	Molecular Formula	Peak Areas Average
Acids	07:05	215.0860	70.0414; 99.0680	Beta-ureidopropionic acid	C_4_H_8_N_2_O_3_	28,087
	07:11	359.0380	204.1133; 289.1786	propanoic acid	C₃H₆O₂	36,531
	08:03	226.1468	211.1231; 226.1468	DTXSID40154910	C_16_H_18_O	1,352,933
	25:44	344.0699	85.0397; 114.0423	CTK9A2446	C_31_H_44_N_4_O_5_	1,360,725
	28:47	268.0383	104.0622; 68.5348	Cannabinolic acid	C_22_H_26_O_4_	3539
Alcohol	04:05	309.3482	57.0700; 125.1325	Henicosanol	C_21_H_44_O	615,961
Aldehyde	05:53	178.1101	69.0574; 178.1101	Benzothiazole, 2-amino-5,6-dimethyl-	C_9_H_10_N_2_S	132,889
	06:28	167.1353	45.0576; 87.0680	3-Cyclopentylpropionamide, N,N-dimethyl-	C_10_H_19_NO	411,894
	08:01	140.0580	111.0554; 140.0580	Benzaldehyde, 4-benzyloxy-3-methoxy-2-nitro-	C_15_H_13_NO_5_	689,665
	04:19	109.0615	109,0887; 43.0887	propan-2-one,	C_6_H_14_O	6,431,210
Alkaloids	30:34	324.1647	204.1126; 323.161	Quinine	C_20_H_24_N_2_O_2_	10,651
Alkene	07:15	270.0476	55.0544; 83.0855	(E)-5-Octadecene	C_18_H_36_	381,486
	11:00	405.0818	57,0699; 97.1013	Nonacosene	C_29_H_58_	164,163
Alkyl-phenylketones	09:24	150.0676	107.0493; 135.0442	1-acetyl-2-hydroxy-5-methylbenzene	C_9_H_10_O_2_	447,852
Amide	07:53	147.1180	59.0367; 126.0312	N-(6-Chloro-2-pyrazinyl)-2-(1-piperidinyl)acetamide	C_11_H_15_ClN_4_O	264,451
Amines	03:15	224.8298	68.0258; 1120714	Cyclopentanoneoxime	C_5_H_9_NO	292,727
	03:23	169.0877	141.0699; 169.0886	4-phenylaniline	C_6_H_5_-C_6_H_4_NH_2_	16,716
	03:45	420.0086	204,1133; 275,1631	Ethyl (1S)-1-phenyl-3,4-dihydro-1H-isoquinoline-2-carboxylate	C_18_H_19_NO_2_	453,344
	04:50	181.0014	148,0217; 181,0014	dimethyl ketone	CH_3_COCH_3_	1,018,170
	06:44	208.0854	70.0652; 97.0889	2-Ethyl(dimethyl)silyloxybutane	C_8_H_20_OSi	48,784
	06:50	157.0885	128.0620; 156.0808	3-methyl-4-phenylpyrrole	C11H1_1_N	45,870
	07:01	287.9991	130.0652; 166.0739	3-Methylbutyl N-heptafluorobutyryltryptophanate	C_12_H_24_O_2_	712,718
	08:40	284.0485	84,0807; 31,0184	N,N,4,4-Tetramethylcholestan-3-amine	C_31_H_57_N	79,967
	29:13	136.0994	108.0684; 135.0917	2,6-Diethylpyrazine	C_8_H_12_N_2_	7879
	11:02	277.2139	85.0523; 177.0658	Stearamide mea,	C_20_H_41_NO_2_	295,552
Amino acids	10:03	223.6640	99.0512; 125.0710	Phenylalanine, methyl ester	C_10_H_13_NO_2_	2,534,272
	09:34	257.1639	154.0652; 171.0918	5-Hydroxy-L-tryptophan	C_11_H_12_N_2_O_3_	17,277,429
Anticholinergic agent	07:56	176.9971	86.0386; 99.0679	Benactizina	C_20_H_25_NO_3_	28,780
Aromatic hydrocarbons	10:19	268.9954	121.0648; 149.0961	4,4’-(1,2-Diethylethylene)bis(anisole)	C_20_H_26_O_2_	113,270
Azole	05:48	241.1339	83.0730; 193.0844	4-Methoxy-6-methyl-1,3,5-triazin-2-amine	C_5_H_8_N_4_O	98,612
Benzene	10:24	117.0574	90.0465; 117.0574	1*H*-indole	C_8_H_7_N	15,162,269
Benzene/amines	29:23	155.0729	127.0542; 155.07729	N-(4-methylphenyl)pyridin-3-amine	C_25_H_21_N	9681
Beta carbolines	07:27	168.0685	140.0498; 184.1125	Carbazoline	C_11_H_8_N_2_	42,951
Carboxamides	07:46	287.9859	59.0368; 83.0855	Oleamide	C_18_H_35_NO	33,866
	03:13	219.2235	152.1435; 31.0185	3-Ethyl-5-methyl-2,4-heptadiene	C_10_H_18_	673,311
	03:29	347.0880	135.0804; 156.0808	Propanone,	C_3_H_6_O	18,547
Cytochrome	30:56	281.1641	85.1013; 149.0231	Tetrahydropyran Z-10-dodecenoate	C_17_H_30_O_3_	18,827
Esters	09:14	389,0513	125.0709; 153.0658	Epoxypropanol methacrylate	C_7_H_10_O_3_	22,299
	27:59	347.3253	135.0903; 1490958	Phthalic acid, di(8-chlorooctyl) ester	C_24_H_36_C_l2_O_4_	10,848
Ether	09:22	136.0128	45.0336; 59.0492	Carbitol,	C_6_H_14_O_3_	234,708
Fatty acids	06:14	181.0346	71,0855,; 98.0966	Undec-2-en-4-ol	C_11_H_22_O	129,187
	10:16	270.2548	74.0362; 143.1068	Palmitic acid methyl ester	C_17_H_34_O_2_	12,371,227
Halogenated pyrroles.	08:32	268.0174	44.133; 59.0367	Phantom	C_15_H_11_BrClF_3_N_2_O	1,251,965
Heterocyclic compound	07:15	221.1369	124.0633; 180.0894	2,3-Dihydrothiophene	C_4_H_6_S	392,746
	29:51	211.1444	70.0652; 154,0740	L-Phe-D-Pro lactam	C_11_H_18_N_2_O_2_	15,224
	08:25	315.1130	91.0544; 300.0896	2,6-Dimethyl-3-isopentylpyrazine	C_17_H_18_ClN_3_O	96,316
Ketone	05:25	246.1358	91,0543; 127,0864	Propanone,	(CH_3_)_2_CO	403,574
Organic acid	06:31	269.9949	179,0688; 263,9868	Cypionic acid	C_20_H_22_O_2_	153,948
Organosilicon compounds	30:17	244.0916	91.0544; 167.0525	Dimethoxydiphenylsilane	C_14_H_16_O_2_Si	55,316
	30:19	211.1355	168.0809; 211.1335	Methyloctyldimethoxysilane	C_10_H_16_OSi	2,187,251
Phenols	05:24	206.1665	57.0700; 191.1430	2,4-Di-tert-butylphenol	C_14_H_22_O	451,861
Polycyclic aromatic hydrocarbon	09:37	202.0774	101.0393; 202.0774	Benzo(jk)Fluoren	C_20_H_12_	44,281
	29:19	202.0777	178.0976; 202.0777	Beta-Pyrene	C_16_H_10_	3558
Protein	05:27	219.1606	91.0543; 99,0555	Cyclo(Ala-Phe)	C_12_H_14_N_2_O_2_	9,331,496
Pyridines	05:45	231.1038	75.0234; 231.1038	2,6-DPhPy	C_17_H_13_N	304,326
	07:33	182.0840	154.0654; 182.0840	brevicolline	C_12_H_10_N_2_	43,227
Pyrrolidine	06:00	180.9439	41.0387; 84.0444	Pyrrolidon	C_4_H_7_NO	35,732
Pyrrolidinone	06:01	190.0754	41.0387; 84.0444	5-(Cyclohexylmethyl)-2-pyrrolidinone	C_11_H_19_NO	50,942
Quinolines	10:32	227.9064	154.0737	Acetone anil	C_12_H_15_N	280,959
Unsaturated aliphatic hydrocarbons	06:35	252.2804	55.0545; 83.0855	3-eicosene	C_20_H_40_	474,139

**Table 5 toxins-14-00712-t005:** Secondary metabolites yielded by *S. saprophyticus* (ATCC 35552).

Metabolites Class	RT (min)	Observed Ion *m/z*	*m/z*	Metabolite Name	Molecular Formula	Peak Areas Average
Acids	25:44	324.1651	44.0497; 74.0237	Succinic acid	C_4_H_6_O_4_	748,584
Acids/Esters	09:33	156.5059	41.0388; 84.0444	2-Pyrrolidinecarboxylic acid-5-oxo-, ethyl ester	C_11_H_14_O_3_	30,578,538
Acids/Propionates	19:00	210.0846	139.0865; 225.0758	Linalyl propionate	C_7_H_11_NO_3_	125,699
Adipates/Acids	21:38	299.2737	59.0367; 129.0547	Diethylhexyl adipate	C_13_H_22_O_2_	1,358,675
Alkanes	12:58	225.4679	57.0700; 71.0855	Cetane	C_22_H_42_O_4_	234,536
Alcohols	04:52	174.0471	74.0237; 86.0964	Undec-2-en-4-ol	C_16_H_34_	1,854,763
	14:06	264.0909	138.0789; 151.0867	2-(4-Fluoro-phenyl)-cyclohexanol	C_11_H_22_O	649,839
	04:19	136.0144	45.0337; 59.0493	ethyl carbitol	C_12_H_15_FO	1,2961,908
Alkaloids/Ergotamines	21:47	567.0431	125.0711; 153.0660	Ergotamine	C_6_H_14_O_3_	41,429,291
Alkaloids	16:15	401.9806	57.0212; 114.0424	Veratramine	C_33_H_37_N_5_O_5_	4,429,604
Alkane	21:34	336.3753	57.0699; 97.1013	Cyclotetracosane	C_27_H_39_NO_2_	4,929,019
Alkene	19:50	266.2966	57.0700; 83.0855	Nonadec-1-ene	C_24_H_48_	7,761,975
	09:24	224.0703	55.0544; 83.0855	Cetene	C_19_H_38_	328,938
	12:52	264.1646	44.0497; 86.0964	Octadec-9-ene	C_16_H_32_	1,546,382
Amides	22:59	364.4082	30.0343; 44.0497	Histidine amide	C_18_H_36_	16,30,440
	21:27	286.2631	59.0367; 72.0444	oleic acid amide	C_6_H_10_N_4_O	14,326,846
	19:48	255.256	59.0367; 72.0444	Cetyl amide	C_18_H_35_NO	2,500,587
Amines	13:13	192.1621	177.1388; 192.1612	2-Propanamine, N-[(3-nitrophenyl)methylene]-	C_16_H_33_NO	61,970
	25:54	393.3375	250.1583; 322.2517	bis(4-t-octylphenyl)amine	C_11_H_14_N_2_O_2_	103,487
	10:19	227.0685	44.0497; 57.0211	N,N,4,4-Tetramethyl-5alpha-cholestan-3beta-amine	C_28_H_43_N	4,308,153
	23:20	264.279	144.0806; 171.0916	Phenoharmane	C_31_H_57_N	134,169
	19:22	238.2169	85.0523; 98.0602	N-lauroylethanolamine	C_18_H_18_N_2_	202,235
Aromatic Hydrocarbons	18:52	202.0776	101.0395; 202.0775	Benzo(jk)fluorene	C_14_H_29_NO_2_	160,110
Aromatic Heterocyclic organic compound	08:17	117.0574	90.0465; 117.0574	Indole	C_16_H_10_	5,054,596
	09:37	131.0729	130.0652; 145.0762	Skatole	C_8_H_7_N	412,362
Azoles/Pyrroles	12:54	157.0886	128.0622; 156.0809	3-Me-4-Ph-pyrrole	C_9_H_9_N	1,455,835
	17:08	221.1279	124.0632; 180.0895	2-undecyl-1H-pyrrole	C_11_H_11_N	2,640,782
Azoles/Triazoles	23:01	315.1132	119.0856; 300.0898	Tinuvin 326	C_15_H_27_N	742,229
	18:50	225.0897	44.0496; 86.0964	Tinuvin P	C_17_H_18_ClN_3_O	817,086
Biphenyl Compounds	12:10	169.0888	115.0544; 169.0888	4-Biphenylamine	C_13_H_11_N_3_O	130,219
Benzoate	11:53	194.0938	211.0285; 149.0599	4-Ethoxy ethylbenzoate	C_12_H_11_N	315,670
Butyl Esters	15:21	219.0888	104.0622; 135.0805	Butyl fumarate	C_12_H_20_O_4_	253,338
Esters	12:00	270.0461	74.0237; 86.0964	isohexyl ester	C_16_H_30_O_4_	118,061
Ethyl Ethers	03:13	125.0713	44.0496; 86.0964	Chloromethyl isobutyl ether	C_3_H_7_ClO	1,473,624
Fatty Acids	17:11	270.2552	30.0343; 74.0237	Methyl palmitate	C_17_H_34_O_2_	358,875
Heterocyclic Compounds/Pyridines	10:54	155.073	127.0543; 155.0730	3-PhenyIpyridine	C_11_H_9_N	1,702,430
Hydrocarbons	17:53	280.3128	83.0856; 97.1014	3-eicosene	C_20_H_40_	4,116,824
Indole	18:13	168.0684	140.0497; 168.0684	Carbazoline	C_11_H_8_N_2_	1,824,128
Indole/Benzene	18:06	182.0839	30.0343; 74.0237	Azobenzene	C_12_H_10_N_2_	2,035,127
Ketones	17:07	225.1517	140.0581; 196.1208	2-Methyl-4-amino-6-methoxy-s-triazine	C_5_H_8_N_4O_	586,977
Nitriles	10:36	154.0527	127.0418; 154.0527	Isoquinaldonitrile	C_10_H_6_N_2_	236,359
	03:18	108.0684	81.0574; 108.0684	1,6-Dihydroimidazole[4,5-d]imadazole/Diaminomaleonitrile	C_8_H_7_N	1,245,557
Olefin/Alkenes	23:09	343.066	57.0701; 97.1012	Nonacosene	C_29_H_58_	2,784,072
Organosilicon Compunds	11:15	506.1064	73.0468; 147.0657	CTK6B0391	C_18_H_52_O_7_Si_7_	1,529,176
	13:58	490.0586	73.0468; 355.0699	Hexadecamethyl-cyclooctasioxane	C_16_H_48_O_8_Si_8_	661,927
Phenols	11:45	206.1666	57.0700; 191.1431	2,4-Di-tert-butylphenol	C_14_H_22_O	1,293,238
Pyrenes	19:24	202.0777	88.0308; 202.0777	ß-Pyrene	C_16_H_10_	143,873
Pyridines	20:27	231.1037	74.0236; 86.0964	Pyridine, 2,6-diphenyl-	C_17_H_13_N	436,201
Pyrroles	04:58	109.0887	94.0653; 109.0887	2,3,4-Trimethylpyrrole	C_7_H_11_N	3,128,380
	04:25	269.0491	59.0367; 151.0898	Pylon	C_15_H_11_BrClF_3_N_2_O	1,066,266
	13:42	157.0887	77.5362; 156.0810	2-Methyl-5-phenylpyrrole	C_11_H_11_N	230,614
Quinazolines/heterocyclic compounds	08:56	144.0683	98.0602; 144.0683	4-Methylquinazoline	C_9_H_8_N_2_	215,987
Sulphides/sulphur compounds	03:59	125.9626	110.9393; 125.9626	Dimethyltrisulfane	C_2_H_6_S_3_	1,926,298
Thiophenes	13:09	169.053	109.0762, 137.0710	benzothiophene sulfone	C_12_H_8_O_2_S	330,702

**Table 6 toxins-14-00712-t006:** The Kruskal Wallis test results.

Test Statistics ^a,b^	
	%Peaks Area Average
Kruskal-Wallis H	1.854
df	2
Asymp. Sig.	0.396

^a^ Kruskal Wallis Test; ^b^ Grouping Variable: Metabolites Class.

**Table 7 toxins-14-00712-t007:** Lethal concentration (LC_50_) of secondary metabolites extracted from *S. aureus*, *S. epidermidis,* and *S. saprophyticus*.

Compound	Origin	Lethal Concentration (LC_50_) in mg/mL)
(1) 4-Methyl-pentyl amine	*S. saprophyticus* (ATCC 35552)	0.0231 ± 0.0027
(2) Fluoranthene	*S. aureus (*isolated from milk	0.0167 ± 0.0003
(3) Cyclo (leucyl-prolyl	*S. aureus (*isolated from milk)	0.0310 ± 0.0012
(4) Oleamide	*S.epiderdis* (ATCC 51625)	0.0333 ± 0.0012
(5) Veratramine	*S. saprophyticus* (ATCC 35552)	0.0274 ± 0.0007
(6) Methyl palmitate	*S. epidermidis* (ATCC 51625)	0.0441 ± 0.0040
(7) 3-methyl-2-phenyl-1H-pyrrole	*S. aureus* (isolated from milk	0.0341 ± 0.0093
(8)1,2,6-Hexanetriol	*S. saprophyticus* (ATCC 35552)	0.0333 ± 0.0031
(9) Succinic acid	*S. saprophyticus* (ATCC 35552)	0.0334 ± 0.0017
Doxorubicin		0.0101 ± 0.0004

**Table 8 toxins-14-00712-t008:** ANOVA table indicating the relation between the % viability of Vero cell and the concentration of the sample.

ANOVA ^a^						
Model		Sum of Squares	df	Mean Square	F	Sig.
1	Regression	2173.098	1	2173.098	136.707	0.000 ^b^
	Residual	826.594	52	15.896		
	Total	2999.693	53			

^a^ Dependent Variable: Percent Viability; ^b^ Predictors: (Constant), Concentration.

**Table 9 toxins-14-00712-t009:** A summary for evaluating the method used in data analysis.

Model Summary ^b^				
Model	R	R Square	Adjusted R Square	Std. Error of the Estimate
1	0.851 ^a^	0.724	0.719	3.98698469

^a^ Predictors: (Constant), Concentration; ^b^ Dependent Variable: Percent Viability.

**Table 10 toxins-14-00712-t010:** Bacterial strains used for the experimental work.

Bacterial Strain	Source/or Supplier of Bacterial Strain	Strain ATCC
*Staphylococcus aureus*	Cow milk, MSA agar confirmed	Isolated from milk
*Staphylococcus epidermidis*	Thermofisher Scientific, South Africa	ATCC 51625
*Staphylococcus saprophyticus*	Thermofisher Scientific, South Africa	ATCC 35552

## Data Availability

Part of the data is available on https://uoj-researchportal.esploro.exlibrisgroup.com/esploro/outputs/doctoral/Secondary-metabolites-produced-by-Staphylococcus-species/9910534007691?institution=27UOJ_INST (accessed on 24 August 2022).

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
