# Peer review of "Investigating the Toxicity of Compounds Yielded by Staphylococci on Vero Cells"

_toxins, 2022, doi:10.3390/toxins14100712_

Round 1

Reviewer 1 Report

This manuscript describes the interesting effect of bacterial secondary metabolites on Vero cells from African green monkeys, presumable as a proxy for human cells.  This is an interesting line of research and I would encourage the authors to persist, however this manuscript has a number of problems which need to be addressed.

1. What is Table 1 for?  It lists a number of bacterial pathogens for which there is no data in the rest of the manuscript.  So to describe them as "Bacterial strains used for experimental work" seems odd to me.  Table 2 lists only secondary metabolites from a selection of staphylococci only.

2. The discussion talks about the actions of secondary metabolites, but there is no data pertaining to antimicrobial activity.  Is this a mistake?  As it stands, the discussion is a bit of a mess and doesn't read coherently.

3. Why were these secondary metabolites chosen?

4. Why are the Vero cells used a good model/substrate for toxicity assays?

4. It would be insightful to have some understanding of the in vivo concentrations of these secondary metabolites, but that is virtually impossible, so what is their concentration in the axenic cultures used in this research?  Concentration is key here because the action of these metabolites on cells could be radically different at lower levels than those used here.  At the very least, this need to be clearly acknowledged, with corresponding examples, in the discussion.  

Author Response

Dear Reviewer

We attended to all the issues you highlighted.

Reviewer 2 Report

The manuscript entitled

 Investigating the toxicity of secondary metabolites yielded by Staphylococci on Vero Cells”

is concise, and provides important informations. However, I have some few comments, which are listed below.

For example:

Abstract: Is unclear and the authors should rephrase the whole abstract to be more easy for understanding and more relevant to the whole manuscript.

Introduction:

Line 24: ESKAPE is an acronym comprising the six highly virulent and antibiotic resistant bacterial pathogens including: Enterococcus faecium, Staphylococcus aureus, Klebsiella pneumoniae, Acinetobacter baumannii, Pseudomonas aeruginosa, and Enterobacter spp. The authors should state this firstly before writing the acronym.

Lines 30 and 31: This sentence should not be included in the introduction; Future studies will focus on metabolites from other bacteria within the ESKAPE group. Please remove.

Lines 32-34: It can be hypothesized that secondary metabolites produced by Staphylococcus species 32 are efficient in exhibiting their antimicrobial activity against ESKAPE bacteria [15]. It can 33 also be hypothesized that secondary metabolites produced by Staphylococcus species will 34 not be efficient against ESKAPE bacteria [13]. This is totally unclear, if the authors mean that metabolites by Staphylococcus spp. can be used as antimicrobials against the same spp., then please state what is the exact species and the exact by-product and against which spp. (Lysostaphin is an example for this) and how could and could not be these byproducts used as antimicrobials at the same time?

Line53: What do the authors mean by an abnormal number of chromosomes?

Materials and Methods:

Line 86: Table1; all references are cited as errors. Please correct

Results:

Line 212: Please correct; consist of or more aromatic.

Overall, the methodology and results must be presented in more clear scentences.

Author Response

(The authors gave the same response as above.)

Reviewer 3 Report

This manuscript describes an attractive topic, the secondary metabolites produced by Staphylococci. However, according to the manuscript content, it is not certain what was analyzed by authors. This manuscript contains essential flaws unfortunately, even though authors identified substances by GC-MS.

1. Authors did not state the meaning, definition of "secondary metabolites". In Methods section. Authors cultured bacteria, and culture broth was processed for the samples. It is not certain whether such samples can be called as "secondary metabolites.  Amount of the substances may be affected by number of bacterial cells in culture, condition of culture, etc. But these factors were not referred. 

2. Description of Method section is unusual. Authors first listed instruments, and chemicals/reagents. Such way of description is not taken usually in scientific articles.

3. List of bacterial strains, Table 1: This is strange. All the Reference columns contains no information. What does "Thermofischer Scientific, South Africa" means in Source column? Were the strains purchased from this company? Why was there a column of "Select virulence gene profile" ? Actually the listed genes have no meaning. In addition, blaCTX-M, blaOXA, blaSHV are not virulence genes.

4. Only S. aureus was derived from cow milk, but not ATCC strain. In such case, readers do not understand the origin and identity of this strain. If this isolate is regarded as an established strain, there must be a strain name, and might be published previously anywhere. However, there is no strain information. In such case, it is completely not sure whether this strain represents general S. aureus isolates.  

Author Response

Dear Reviewer

We attended to all the issues highlighted.

Round 2

Reviewer 3 Report

The revised manuscript appears to be considerably improved. However, only one point should be clarified and revised.

line 97- "The S. aureus 97 sample identified as ATCC 29213 was obtained from cows with clinical features of mastitis": meaning of this sentence cannot be understood. ATCC29213 is only one unique S. aureus strain provided by ATCC. This strain was isolated from wound, according to ATCC information. Authors' S. aureus samples cannot be identified as ATCC29213. Probably authors intended another meaning. Anyhow this sentence is NOT correct, thus must be rephrased.   Table 1 should be also corrected, for description "Isolated from milk and identified as ATCC 29213". This reviewer thinks that authors might misunderstand this point and added ATCC according to reviewers' comments. If authors used only S. aureus isolates from milk, not standard strain, it should be written as it is. In such case, any information of characteristic (antimicrobial susceptibility, genotype, presence of toxin genes, etc.) to these strains should be added. Otherwise, readers cannot understand whether the S. aureus isolates were somewhat special ones or pathogenic ones, otherwise ordinary isolates.

Author Response

Dear Reviewer

We tried to address the concerns that you mentioned. All the changes in the methods, results and discussion sections are written in red
